# Early anthropoid femora reveal divergent adaptive trajectories in catarrhine hind-limb evolution

Sergio Almécija [1,2,3]*, Melissa Tallman [4], Hesham M. Sallam [5], John G. Fleagle [6], Ashley S. Hammond[1,2] & Erik R. Seiffert [7]

The divergence of crown catarrhines—i.e., the split of cercopithecoids (Old World monkeys) from hominoids (apes and humans)—is a poorly understood phase in our shared evolutionary history with other primates. The two groups differ in the anatomy of the hip joint, a pattern that has been linked to their locomotor strategies: relatively restricted motion in cercopithecoids vs. more eclectic movements in hominoids. Here we take advantage of the first well-preserved proximal femur of the early Oligocene stem catarrhine *Aegyptopithecus* to investigate the evolution of this anatomical region using 3D morphometric and phylogenetically-informed evolutionary analyses. Our analyses reveal that cercopithecoids and hominoids have undergone divergent evolutionary transformations of the proximal femur from a similar ancestral morphology that is not seen in any living anthropoid, but is preserved in *Aegyptopithecus*, stem platyrrhines, and stem cercopithecoids. These results highlight the relevance of fossil evidence for illuminating key adaptive shifts in primate evolution.

[1] Division of Anthropology, American Museum of Natural History, Central Park West at 79th Street, New York, NY 10024, USA. [2] New York Consortium in Evolutionary Primatology, New York, NY, USA. [3] Institut Català de Paleontologia Miquel Crusafont, Universitat Autònoma de Barcelona, c/ Columnes s/n, Campus de la UAB, 08193 Cerdanyola del Vallès, Barcelona, Spain. [4] Department of Biomedical Sciences, Grand Valley State University, 1 Campus Drive, Allendale, MI 49401, USA. [5] Mansoura University Vertebrate Paleontology Center, Department of Geology, Faculty of Science, Mansoura University, Mansoura 35516, Egypt. [6] Department of Anatomical Sciences, Health Sciences Center, Stony Brook University, Stony Brook, NY 11794-8081, USA. [7] Department of Integrative Anatomical Sciences, Keck School of Medicine, University of Southern California, Los Angeles, CA, USA. *email: salmecija@amnh.org

Extant catarrhine primates comprise cercopithecoids (Old World monkeys (OWMs)) and hominoids (apes and humans). Owing to the patchy fossil record documenting the very early stages of the two crown groups, their estimated divergence during the early or late Oligocene has long been one of the more mysterious phases in catarrhine evolution, e.g. refs. [1,2]. Modern cercopithecoids retain a more generally primitive (i.e. plesiomorphic) overall body plan than do living hominoids, the latter of which exhibit postcranial adaptations that suggest an orthograde arboreal ancestry[3–6]. However, other specific aspects of their morphology (e.g. dentition, elbow) are more specialized (i.e. autapomorphic) relative to stem catarrhines[7–10]. Understanding the sequence of morphological changes in catarrhine, and especially hominoid, evolution is complicated by fossil taxa that exhibit mosaic morphologies that are not seen in any living species[11–14]. For example, palaeontological evidence suggests that some Miocene hominids (i.e. the great ape and human clade) exhibited orthograde adaptations without the accompanying specialized features related to below-branch suspensory adaptations that are seen in some living hominoids[15–17].

Anatomically, the primate hip complex represents a key region of study as it differs substantially between modern catarrhine clades—cercopithecoids and hominoids—and has been related to their different specialized locomotor strategies: stereotyped in the former vs. more eclectic in hominoids (in relation to their specialized antipronograde arboreal locomotion)[5,18,19]. Specifically for the case of the hip, although Miocene hominoids exhibit proximal femora that are similar to those of living hominoids[18,20,21], the complementary side of the hip joint (the pelvis) was more 'monkey-like' in morphology[20,22,23], suggesting that the proximal femoral morphology of early hominoids could represent the plesiomorphic condition for catarrhines, e.g. ref. [20]. This hypothesis has deep evolutionary implications, as there is kinesiological evidence showing that the hominoid proximal femur allows for enhanced range of motion at the hip joint, e.g. refs. [18,24,25], which facilitates the array of specialized arboreal postures observed in living apes. If hominoids are indeed more 'primitive' than crown OWMs for the particular case of the proximal femur, this would imply that the locomotor repertoire of stem catarrhines and stem OWMs might have been characterized by far more eclectic locomotor behaviours than previously envisioned.

This study investigates the evolution of the catarrhine hip complex (from the proximal femoral side) since the Oligocene and the implications for the locomotor capabilities of the cercopithecoid–hominoid ancestor. To accomplish this goal, we use a series of stepwise analyses, combining three-dimensional geometric morphometrics (3DGM; see Supplementary Table 1) and evolutionary modelling within a multi-regime, multivariate framework across a large sample of living and fossil anthropoid femora (Supplementary Tables 2 and 3). Importantly, relevant fossils are incorporated into the analyses, including a new femur (DPC 24466; Fig. 1) of *Aegyptopithecus zeuxis*, an early Oligocene advanced stem catarrhine from Egypt that can uniquely inform this question[26]. This fossil is key because previously known *Aegyptopithecus* femora do not preserve enough of the proximal region to test hypotheses about the evolution of hip mobility in stem catarrhines[27]. Given that *Aegyptopithecus* is close in age to the predicted divergence of cercopithecoids and hominoids[1] and is widely accepted as an advanced stem catarrhine[2], we proceed with the assumption that this taxon is more likely than not to closely approximate the morphology of the last common ancestor of cercopithecoids and hominoids and is not already highly autapomorphic.

The results of this study show that cercopithecoids and hominoids have undergone divergent evolution of the hip complex (on the femoral side) from an ancestral morphology not present in living anthropoids, which is represented in *Aegyptopithecus*, stem platyrrhines, and stem cercopithecoids. Based on the 'intermediate' morphometric position identified for *Aegyptopithecus* and its inferred plesiomorphic evolutionary regime, the ancestor of cercopithecoids and hominoids was not specialized towards either of the distinct locomotor repertoires exhibited by modern groups. These results highlight the relevance of fossil evidence for illuminating key adaptive shifts in primate evolution.

## Results

**The new *Aegyptopithecus* femur.** DPC 24466 (Fig. 1c) was found in 2009 at Quarry M, in the upper sequence of the Jebel

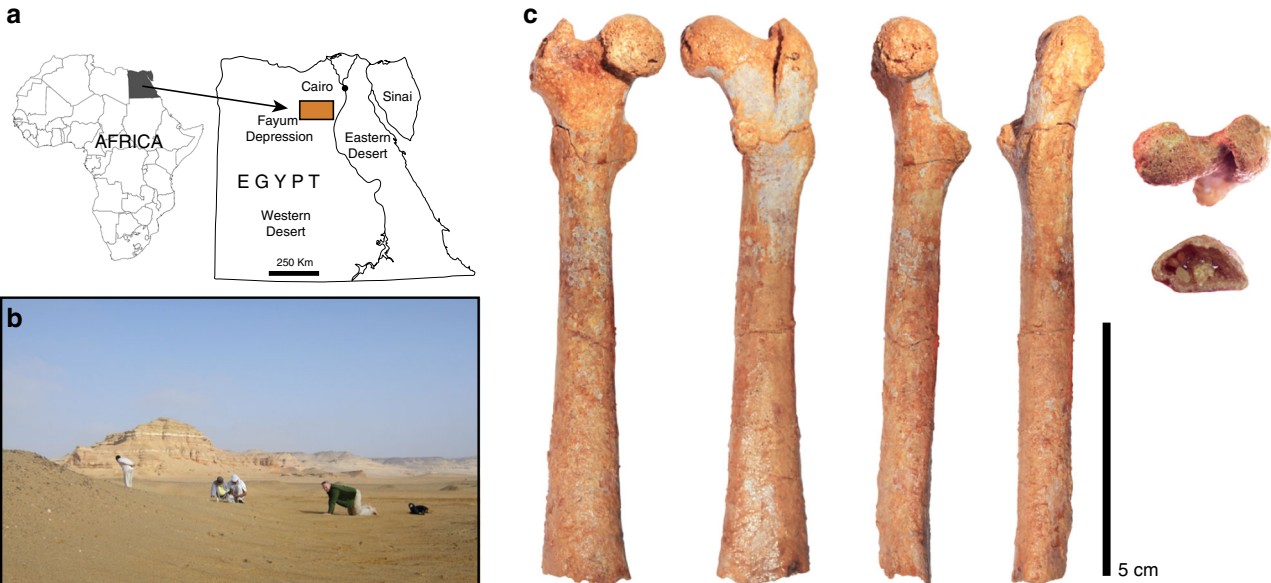

**Fig. 1** Site location and views of the new *Aegyptopithecus* femur (DPC 24466). **a** Location of the Fayum Depression within Egypt, where the Quarry M locality is. **b** Detail picture of the Quarry M locality during its survey in 2009 (photo credit: Mark Mathison). **c** Anterior, posterior, medial, lateral, proximal and distal views, respectively, of DPC 24466

Qatrani Formation (Fig. 1b). This level is considered to be 29.5–30.2 Ma based on the preferred magnetostratigraphic correlation of Seiffert[28] but could be even younger if Gingerich[29] and Underwood et al.[30] are correct in placing the entire Jebel Qatrani Formation within the Oligocene. *A. zeuxis* is therefore at least ~4.5–5.0 Ma older than the oldest known fossil hominoids and cercopithecoids, which have been recovered from a 25.2 Ma site in the Nsungwe Formation of Tanzania[1]. The attribution of DPC 24466 to *A. zeuxis* is based on the fact that there are no other primate taxa in the same size range identified at Quarry M (or the nearby Quarry I) after over five decades of intensive fieldwork at the site[31], during which thousands of vertebrate fossils have been identified. Simons[31] considered it likely that the first specimen of *A. zeuxis*, collected by George Olsen in 1907 (and misidentified as a 'carnivore'), was found at or very close to Quarry M, indicating that the quarry has potentially been known for 112 years. Like several other sites in the upper sequence of the Jebel Qatrani Formation, Quarry M is a coarse-grained fluvial site that was historically 'wind-harvested' by sweeping away the *serir* (desert pavement) each year, thereby allowing the unconsolidated sand and gravel to blow away and reveal fossils that are embedded in underlying sediments.

DPC 24466 is a right femur that is essentially complete except for the distal portion, which is broken off at the shaft proximal to the epicondyles. The preserved length of the specimen is 112.1 mm, and based on the distal mediolateral expansion of the shaft (proximal to the epicondyles), it appears to have been fairly short. The superoinferior diameter of the slightly eroded femoral head is 13.2 mm, and its fovea capitis is posteroinferior to the centre of the articular surface. The superior and posterior portions of the femoral head are continuous with the femoral neck. The articular surface of the head seems to extend onto the femoral neck superiorly and especially posteriorly (although the eroded surface precludes a clear assessment of its full extension), suggesting an emphasis on flexed hip postures, such as those that are used during quadrupedal leaping and running[18,32,33]. The tubercle on the posterior aspect of the femoral neck, sometimes referred to as the paratrochanteric crest[34], which is typically seen in early and middle Miocene hominoids and some living and extinct anthropoids[34–36], is not present. The neck is anteverted <15° and the femoral neck-shaft angle is 125° (in the 'generalized range', i.e. below suspensory primates, consistent with previous estimates; see Supplementary Table 4). The greater trochanter is situated at about the same level as the head or slightly below. The lesser trochanter is well developed and projects from the shaft posteromedially at around 40°, well within the anthropoid range[34]. As previously noted[27], the *Aegyptopithecus* femur exhibits a marked third trochanter, which is a plesiomorphic character present in crown strepsirrhines, adapiforms, omomyiforms, stem anthropoids, stem platyrrhines, stem catarrhines, and some Miocene hominoids[34,36,37].

Most of the femoral shaft is well preserved, and it is relatively straight in all views. The femur is slightly platymeric near the estimated midshaft (Supplementary Fig. 2) and becomes more platymeric distally. It is possible that some degree of post-mortem deformation is contributing to the extreme platymeric appearance of the distal end. The posterior proximal portion of the shaft preserves a keel that most likely corresponds to the insertion of the adductor musculature[38]. Distal to the midshaft, the keel bifurcates into ridges that run to the medial and lateral sides of the distal shaft. A full description of DPC 24466 is available in Supplementary Note 1.

The body mass of the *Aegyptopithecus* individual to which the DPC 24466 femur belonged was estimated using different regressions (hominoid sample, cercopithecoid sample, pooled sample) based on femoral head superoinferior diameter and femoral shaft anteroposterior diameter. Regressions are described in Ruff[39], and the full set of estimates (including 95% confidence intervals) are presented in Supplementary Table 5. Using the three different regressions, the ranges obtained using the femoral head (3.1–5 kg) were less than those obtained using the femoral shaft (6.8–8.5 kg). The body mass estimates using these two proxies probably represent the extreme estimates given that the lower estimate is derived from a slightly abraded femoral head surface, and the larger estimate is derived from a very platymeric shaft (especially distally). Given the evidence presented above, an estimate intermediate between the two extremes is considered as the most reasonable. If this were the case, the estimated body mass range for this specimen is consistent with most published estimates for this taxon, which place *Aegyptopithecus* between 6 and 7 kg[27,40,41].

**Femoral shape variation in anthropoids**. The 3D shape affinities of the living and fossil sample were assessed through 14 3D surface landmarks capturing the overall shape of this part of the bone (Supplementary Fig. 1, Supplementary Table 1). These landmarks were also collected on a large sample of extant and fossil anthropoids (Supplementary Tables 2 and 3, respectively). Specifically, a morphospace summarizing anthropoid femoral shape variation was constructed using the first two axes of a principal components analysis (PCA) carried out on the Procrustes coordinates of the extant species means and fossil individuals, and individual extant specimens were then plotted into the morphospace post hoc (Fig. 2, Supplementary Fig. 3; see 'Geometric morphometrics' in the 'Methods' section). This approach ('between-group PCA' or bgPCA) maximizes variation among groups identified a priori, while accounting for intraspecific variation[42]. When the two first components are inspected together, platyrrhines (or New World monkeys (NWM)), cercopithecoids, and hominoids are separated from each other. Inferred shape changes differentiating the femora of hominoids–cercopithecoids (bgPC1) and platyrrhines–catarrhines (bgPC2) are visually presented in Supplementary Fig. 4 (see more details in the section 'Evolutionary modelling' below). *Aegyptopithecus* (DPC 24466), the early Miocene NWM *Homunculus* (MACN-A 5758), and the middle Miocene stem OWM *Victoriapithecus* (KNM-MB 35518) fall outside the variation of modern taxa, whereas fossil apes *Epipliopithecus* and fossil hominins (i.e. the human clade) fall within the range of modern hominoids. A similar morphospace was constructed in which individual fossils did not contribute to the eigenanalysis but rather were plotted a posteriori (Supplementary Fig. 5). This approach allowed us to contextualize the shape affinities of fossil taxa given only the morphospace defined by living anthropoids. This analysis produced even better separation among the major living clades, while fossil specimens appear closer to each other.

**Evolutionary modelling**. Adaptive patterns of anthropoid femoral evolution were studied using a multi-regime Ornstein–Uhlenbeck (OU) stabilizing selection model[43]. This evolutionary model inspects how different clades undergo shifts towards different phenotypes ('optimal phenotypic values' or 'adaptive peaks') by identifying the different evolutionary 'regimes' (see 'Evolutionary modelling' in the 'Methods' section for details). This 'surface' method[44] was first applied to 'naively' identify possible regime shifts by fitting a series of stabilizing selection models and using a stepwise algorithm to locate phenotypic shifts on a phylogenetic tree (i.e. without previous identification of regimes).

Specifically, starting with an OU model in which all species are attracted to a single adaptive optimum in morphospace, 'surface'

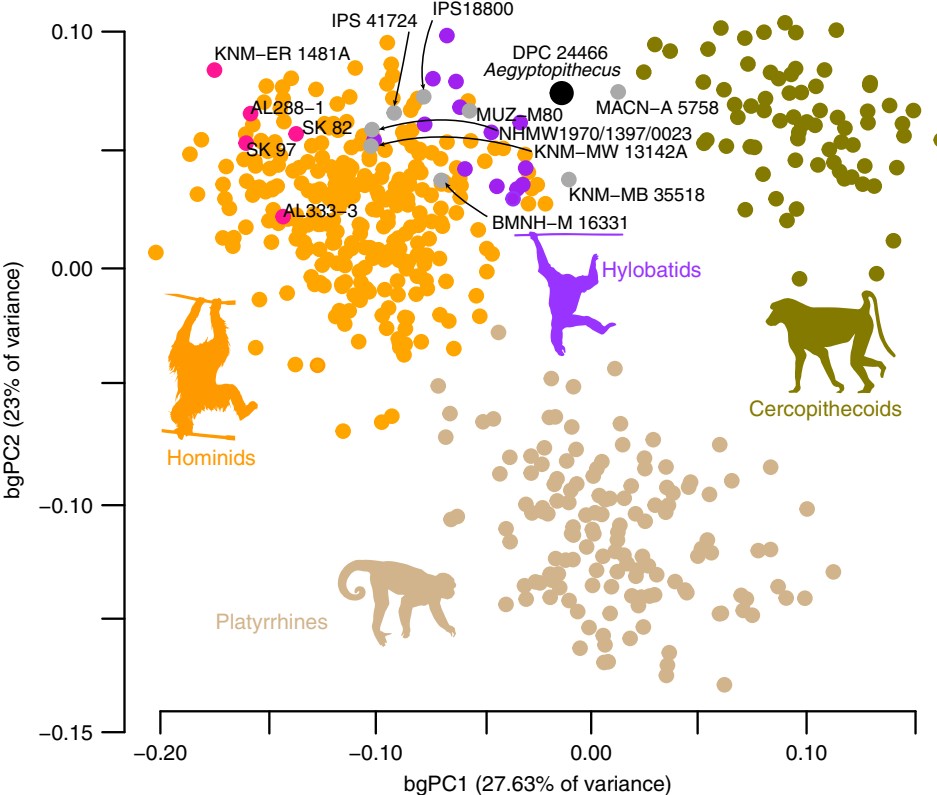

**Fig. 2** Shape analysis of the anthropoid proximal femur. The plot shows the first two principal components of an analysis carried out on the between-group covariance matrix (bgPCA). The groups represent extant species and fossil centroids, with individual specimens and fossils plotted post hoc. Thin-plate-spline (TPS) warped versions of DPC 24466 depicting extremes of variation along each axis are represented in Supplementary Fig 4. The colour codes are as follows: New World monkeys, light brown; Old World monkeys, green; great apes and humans, orange; hylobatids, purple; fossil hominins, pink; other fossil primates, grey; the *Aegyptopithecus* DPC 24466 femur is black. Taxonomic attributions of the fossils represented are: DPC 24466, *Aegyptopithecus zeuxis*; MACN-A 5758, *Homunculus patagonicus*; KNM-MB 35518, *Victoriapithecus macinnesi*; NHMW1970/1397/0023, *Epipliopithecus vindobonensis*; MUZ-M80, *Morotopithecus bishopi*; KNM-MW 13142A, *Ekembo nyanzae*; BMNH-M 16331, *Equatorius africanus*; IPS41724, cf. *Dryopithecus fontani*; IPS18800, *Hispanopithecus laietanus*; AL333-3 and AL288-1, *Australopithecus afarensis*; SK 82 and SK 97, cf. *Paranthropus robustus*; KNM-ER 1481, cf. *Homo erectus*. Silhouettes for *Pongo* and *Symphalangus* were custom made. Silhouette for *Papio* was downloaded from www.phylopic.org and is licensed for free use in the Public Domain without copyright. Silhouette for *Cebus apella* was also downloaded from www.phylopic.org (credit to Sarah Werning, and available for use under CC BY 3.0 license). The authors modified the original colours. Source data are provided as a Source Data file

uses a stepwise model selection procedure based on the finite-samples Akaike information criterion (AICc)[45,46] to fit increasingly complex multi-regime models. At each step, a new regime shift is added to the branch of the phylogeny that most improves model fit across all the variables inspected, and shifts are added until no further improvement is achieved. To verify true convergence, this method then evaluates if the AICc score is further improved by allowing different species to shift towards shared adaptive regimes rather than requiring each one to occupy its own peak. In general, OU modelling is useful to identify potential adaptive regimes and regime shifts, although it is very complex statistically and therefore the results should be interpreted carefully[47].

When this method was applied to the major components of proximal femoral form variation—including extant and fossil species in the eigenanalysis (Supplementary Fig. 3) plus the centroid size (CS; using natural logarithm)—it detected ten different evolutionary regimes acting during anthropoid femoral evolution. Each of the regimes is identified with different colours along the edges of a phylogenetic tree (Fig. 3a) and two representations (subspaces) of the same morphospace showing the inferred adaptive optima (Fig. 3b, c). Representatives of each evolutionary regime are depicted in Fig. 3d and inferred shape

changes associated to each PC axis in Fig. 3b, c are presented in Supplementary Fig. 4. Note that individual data (Fig. 2) and species means/adaptive optima (Fig. 3b) are represented in the same morphospace (their eigenanalysis is exactly the same). *Aegyptopithecus*, *Victoriapithecus*, and *Homunculus* together occupy an anthropoid plesiomorphic regime (red) that is different from any other analysed living or fossil species. Living platyrrhines in the study are inferred to have evolved under four different regimes, the most plesiomorphic of which is represented by *Callicebus* and *Aotus* in different parts of the platyrrhine tree. The *Pithecia–Chiropotes*, ateline and *Cebus–Saimiri* clades represent their own evolutionary regimes, respectively. For catarrhines, all cercopithecines and *Nasalis* share a common regime, whereas *Colobus–Piliocolobus* depart onto their own regime. The differences between both regimes are related to slight differences along PC3 and especially in femoral size (Fig. 3c). Within hominoids, there are three different regimes: All fossil apes and hylobatids, living great apes (convergent regime), and hominins, respectively.

Starting from an *Aegyptopithecus*-like morphology, hominoids and cercopithecoids have evolved in opposite directions along the major axis of phenotypic variation (i.e. PC1 in Fig. 3b): While the former evolved proximally wider femora, with longer anatomical

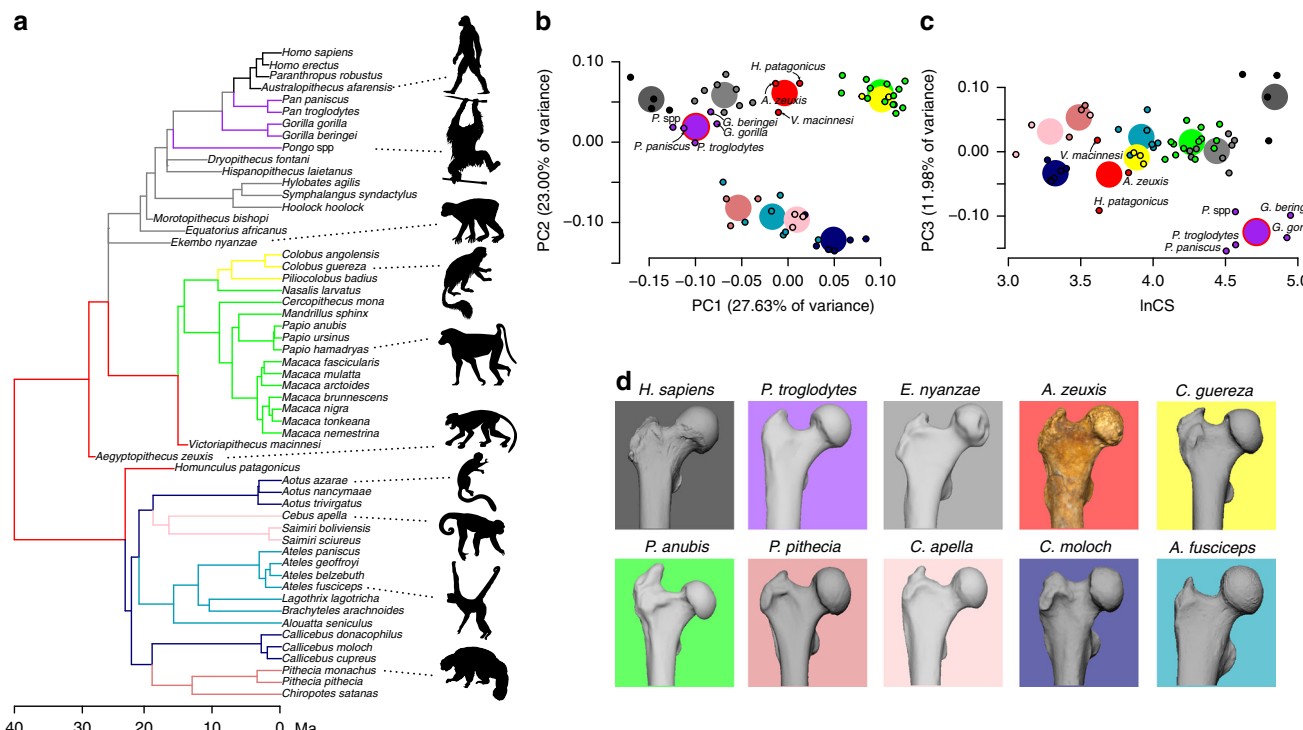

**Fig. 3** Adaptive regimes in the evolution of anthropoid proximal femur form. **a** Time-calibrated chronometric tree depicting the estimated phylogenetic history of adaptive peak shifts during anthropoid proximal femoral shape evolution (each colour represents a different evolutionary regime). **b**, **c** Morphospaces showing the estimated adaptive optima (large circles) and species (small circles) evolving under each evolutionary regime in **a**. A convergent optimum is marked with a red outline. The shape changes associated with each axis are similar to those depicted in Fig. 2 (although with a reversed PC2). Supplementary Fig. 4 depicts the evolutionary history of catarrhine femoral differentiation along the first three PC axes. **d** Morphological comparison of selected femora from each of the evolutionary regimes (scaled to similar mediolateral size). The silhouettes of the primates in **a** illustrate the selected primates of each regime. They were downloaded from www.phylopic.org and are licensed for free use in the Public Domain without copyright. Exceptions are the silhouettes for *Cebus apella* (credit to Sarah Werning and available for use under CC BY 3.0 license), *Ekembo nyanze* (credit to Nobu Tamura and modified by T. Michael Keesey, under CC BY-SA 3.0 license), *Aegyptopithecus zeuxis* (credit to Mateus Zica and modified by T. Michael Keesey, under CC BY-SA 3.0 license). The silhouette of *Pongo* was custom made. Source data are provided as a Source Data file

necks and larger, more proximally situated heads (relative to the greater trochanter), the proximal femora of the latter became narrower, with shorter necks and relatively smaller heads, situated below an enlarged third trochanter (Supplementary Fig. 4). The 'surface' analysis detected a single convergent regime in the tree for living great apes, implying that they independently shifted from a plesiomorphic hominoid regime towards a shared convergent optimum.

Two sensitivity analyses were conducted to verify this result. First, to test the pattern of homoplastic evolution detected for great apes, the analysis was repeated with three alternative phylogenetic trees in which the phylogenetically contentious European fossil great apes represented stem hominids ('tree1'), stem pongines ('tree2'), and stem African apes ('tree3'), respectively. In all cases, the results were identical (Supplementary Fig. 6; also see also 'Phylogenetic tree building' in the 'Methods' section and Nexus trees available as Supplementary Data 1–3 of this article). Second, the statistical fit of this rather complex evolutionary model (i.e. the 'surface' output with 10 different evolutionary regimes) was compared with two more simple models: Brownian motion and a single regime OU model, both showing much less support ($\Delta$AICc > 120; Supplementary Fig. 7).

One of the evolutionary parameters estimated in OU models and specifically by 'surface' is the 'phylogenetic half-life' ($t_{1/2}$), which provides an estimate of the rate of adaptation[43]. It represents the average amount of time that it takes to evolve

half-way to the new optimum given a starting evolutionary regime shift. $t_{1/2}$ estimates for each variable were: $t_{1/2}$ PC1 = 0.561, $t_{1/2}$ PC2 = 1.041, $t_{1/2}$ PC3 = 0.308, and $t_{1/2}$ lnCS = 2.339. This means, for example, that it takes ~0.5 million years for the average anthropoid in the sample to evolve half-way towards a new optimum along PC1, ~1 million years along PC2, ~0.3 million years along PC3, and ~2.3 million years in terms of lnCS. These results could indicate, among other things, that femoral morphological adaptations that distinguish hominoids and cercopithecoids (captured by PC1) occurred twice as fast as the catarrhine–platyrrhine differentiation, which is captured by PC2 (see Figs. 2 and 3 and Supplementary Fig. 4). Changes between great apes and humans (as captured by PC3) occurred more recently and even faster and changes in femoral size (as approximated by lnCS) were the slowest.

A possible caveat of this approach is that 'surface' assumes the input variables are independent from each other to compute the summary of the final AICc scores to choose the best overall model. Adams and Collyer[48] showed that assuming independence among trait dimensions (when they are not independent) can lead to model misspecification using this method (see also refs. 49,50). To address this key potential issue, which is more likely to occur using a phylogenetic PCA than in this case[51], we tested whether individuals' scores for the principal components included in the evolutionary analyses (PCs 1–3) were correlated with each other using phylogenetic generalized least squares. The results indicate that, although PC1 scores are correlated with PC2

and PC3 scores (but not PC2 scores with PC3 scores), the relationship explains only a very small portion of the predictable covariation ($r^2$ PC1–2 = 0.009, $r^2$ PC1–3 = 0.002). In light of this, we are confident that our multivariate multi-OU modelling is robust, especially because this method performs much better using at least 2–4 variables[44]. To complement these results, individual model fittings for each original variable are also reported and discussed in Supplementary Fig. 8.

**Size and phylogenetic signatures**. The relationship between the overall femoral shape and size was inspected by means of phylogenetic multivariate regression[52] of all the Procrustes coordinates and their CS. This relationship turned out to not be statistically significant (permutation test; 1000 rounds). Negative results were also obtained when inspecting the first three axes of variation individually. Therefore, it can be concluded that femoral shape differences among the studied taxa cannot be merely explained by differences in femoral size.

Phylogenetic signal in the anthropoid proximal femur was investigated in both its size (both CS and log-transformed CS, using natural logarithms) and shape (first three PCs and all Procrustes coordinates) using a generalized version of Blomberg's $K$ statistic[53] adapted for high-dimensional data[54]. Using this method, values of $K$ range from $0 \rightarrow \infty$, with an expected value of 1.0 under Brownian motion. Values of $K < 1.0$ describe data with less phylogenetic signal than expected, and values of $K > 1.0$ describe data with greater phylogenetic signal than expected. Using either lnCS or CS, the observed phylogenetic signal is larger than expected in proximal femoral size ($K = 2.410$ and $K = 2.066$, respectively). When the three first PC axes investigated in the evolutionary modelling are inspected at once, proximal femoral shape variation is found to have evolved close to the Brownian expectation ($K = 0.972$), whereas it is less than expected in its overall shape (i.e. all the Procrustes coordinates or all PCs; $K = 0.506$). Finally, when the same array of variables used in the 'surface' analysis are investigated at once, (i.e. the first three PC axes and the lnCS), $K = 2.245$. In all cases, $K$ values were significant ($P \leq 0.001$). $K$ was evaluated statistically via permutation (1000 rounds), where data at the tips of the phylogeny were randomized relative to the tree, and random values of $K$ were obtained for each round and then compared with the $K$ found with the actual tree. These specific results were obtained using tree 1, but equivalent results were found with trees 2 and 3 (see Supplementary Fig. 6).

## Discussion

The results of this study add to the body of knowledge about the morphology and locomotor behaviour of the stem catarrhine *A. zeuxis*[26] and inform our view on the cercopithecoid–hominoid ancestor. As a 'forerunner of apes and humans' (p. 273 in ref. [55]), *Aegyptopithecus* provides a unique window into the nature of advanced stem catarrhines, and, surprisingly—given its similarity to the stem platyrrhine *Homunculus*—even the primitive condition for crown Anthropoidea. *Aegyptopithecus*' body presents a complex mix of primitive and derived features, informing us about a 6–7 kg animal with a plesiomorphic cranial profile and long snout, as is seen in some early hominoids. Its humerus has been described as combining a strepsirrhine-like proximal portion with an *Alouatta*-like distal end[8,56], as well as *Alouatta*-like features of the ulna[57–60] and some aspects of the metatarsals and phalanges[60–62]. Other aspects of the foot morphology have been described as exhibiting 'prosimian' and even Miocene hominoid affinities[63]; the multivariate analysis of Seiffert and Simons[64] found the astragalus of *Aegyptopithecus* to be most similar to those of living and extinct hominoids. In general, based on all

available evidence, *Aegyptopithecus* is best reconstructed as a cautious above-branch arboreal quadruped incorporating some climbing and leaping behaviours, as supported in the past by other evidence from less complete femora[27].

The incorporation of this new *Aegyptopithecus* femur specimen into a 3D morphometric and evolutionary modelling framework further allows us to conclude that the ancestral femoral morphology from which both hominoids and cercopithecoids diverged is not represented by any extant species in the sample, including living platyrrhines (Fig. 3). *Aegyptopithecus* (stem catarrhine) and *Homunculus* (stem platyrrhine) are placed under the same evolutionary regime despite the long time after their inferred evolutionary split, >10 Ma (Fig. 3; see Supplementary Fig. 7 for comparisons of plesiomorphic anthropoid femora). The proximal femoral morphology seen in early hominoids is, to some extent, evidently retained in hylobatids, whereas the three great ape genera share a similar morphology that could have arisen independently (Fig. 3), irrespective of the phylogenetic position of the European fossil great apes (Supplementary Fig. 5). These results support those of previous analyses with a more restricted sample of species and anatomical landmarks[21]. Future discoveries of African late Miocene great apes will better inform the ape–human divergence. Both the femora of the stem cercopithecoid *Victoriapithecus* and the stem platyrrhine *Homunculus* are inferred to have been evolving under the plesiomorphic regime detected for *Aegyptopithecus*. These results suggest that both crown cercopithecoids and hominoids have evolved away from an ancestral catarrhine morphology that differed little from that of the last common ancestor of all Anthropoidea. As all crown platyrrhines occupy a regime that is different from that of the stem platyrrhine *Homunculus*, we infer that some sort of a locomotor shift likely occurred along the terminal part of the platyrrhine stem lineage from the ancestral anthropoid regime retained by *Homunculus* to one that is occupied today by *Aotus* and *Callicebus*. Future studies including more basal stem catarrhines and stem platyrrhines will help to further refine and test this adaptive scenario and the inferred 'starting point' in the evolution of the anthropoid proximal femur.

Regarding the evolution of hip morphology within catarrhines, starting from an *Aegyptopithecus*-like morphology, hominoids and cercopithecoids have evolved in opposite directions along the major axis of phenotypic variation, and, based on the $t_{1/2}$ results, adapted at a faster rate to the new selective regime than the rate of adaptation that separated catarrhines and platyrrhines (i.e. PC1 in Figs. 2 and 3b): While the former evolved proximally wider femora, with longer anatomical necks and larger, more cranially situated heads (relative to the greater trochanter), the proximal femora of the latter became narrower, with shorter necks and relatively smaller heads, situated below an enlarged third trochanter (Supplementary Fig. 4). These different morphologies have been related to enhanced hip mobility (especially abduction) in hominoids vs. enhanced stereotyped flexion–extension along the same axis in cercopithecoids[18,20,24,25,33].

Regarding the phylogenetic signal results, the shape components investigated (PC1–3) show a Brownian phylogenetic signal ($K \sim 1$), which becomes a strong phylogenetic pattern when femoral size is taken into account (lnCS alone and when combined with PC1–3; in either case, $K > 2$). The phylogenetic patterning among large clades (i.e. platyrrhines, cercopithecoids and hominoids) is evident in the morphospaces depicted in Figs. 2 and 3b and Supplementary Fig. 5. However, the phylogenetic signal becomes obscured when all aspects of shape are considered at once (i.e. all Procrustes coordinates or all PCs; $K < 1$). A possible interpretation of these results is that, in high-dimensional data matrices (such as in the case of geometric morphometric data), most of the biologically relevant information is

concentrated in few dimensions or combinations of them (such as the PCs), whereas the rest becomes 'noise' (e.g. see Supplementary Fig. 3).

The results of this study further illuminate key biological aspects of the hominoid–cercopithecoid ancestor and the selective forces that drove the divergence of both crown groups. For example, regarding the hip complex, this ancestor was not specialized towards any of the two trends observed today: neither more stereotyped cursorial and leaping locomotion in cercopithecoids nor specialized arboreal locomotion and enhanced antipronogrady in hominoids[19,65]. Based on the 'intermediate' morphology of the plesiomorphic regime inferred for *Aegyptopithecus*, it might have shown intermediate ranges of hip motion (see ref. [24]), thus compatible with its overall *Alouatta*-like inferred morphology (i.e. a very versatile but cautiously moving 'monkey').

This study also provides new and interesting information about the evolution of pliopithecoids. They constitute a fossil catarrhine group from the Eurasian Miocene with a highly disputed phylogenetic placement: while they are generally considered as stem catarrhines, e.g. refs. [1,66], others consider them as very basal hominoids, e.g. ref. [67]. The position of the *Epipliopithecus* specimen in the proximal femoral morphospace (within hominoids, see Fig. 3) suggests that (a) some pliopithecoids could be hominoids or (b) some pliopithecoids show convergent morphologies with hominoids for the proximal femur. With regard to the evolution of cercopithecoids, these results suggest that extant African colobines are more specialized than other cercopithecoids, matching other anatomical regions such as the hand[68]. However, a larger colobine sample is necessary to explicitly test this hypothesis.

Finally, the results of this study add to the emerging picture, based on different anatomical regions of *Aegyptopithecus*, as well those of early cercopithecoids and early hominoids and hominids, e.g. refs. [15,16,20,23,69,70], that the catarrhine postcranium evolved in a mosaic fashion. Thus, although some extant species could better approximate ancestral morphologies than others for specific anatomical regions, 'overall ancestral body forms' are difficult to assess without thorough investigation of the fossil record. In other words, incorporating palaeontological data into the frameworks provided by modern evolutionary modelling is essential for reconstructing key adaptive shifts in deep time.

## Methods

**Provenance and deposition of DPC 24466**. The specimen was found in 2009 at Quarry M, in the upper sequence of the Jebel Qatrani Formation (Egypt). Its final deposition is the Duke Lemur Center Division of Fossil Primates.

**Geometric morphometrics**. Shape data were obtained from raw coordinates through a full (generalized) Procrustes fit analysis—which rotates, translates and size-scales the landmark configurations to unit of CS—and posterior orthogonal projection onto the tangent space[71]. Subsequently, major patterns of shape variation in the proximal femur among extant anthropoid species were inspected through PCAs on the Procrustes-aligned coordinates of the extant species mean configurations (i.e. the eigenanalysis is carried out on the species means) using the covariance matrix. Intraspecific variation of the extant samples, PCA scores for all the original individuals were computed a posteriori using vector products. The method—also called 'between-group PCA'—is extensively explained elsewhere[42]. Shape changes along the PC axes were computed by warping (using thin-plate-spline (TPS) deformation) a single 3D model of the DPC 24466 femur along the different PC axes[72].

All morphometric analyses, including multivariate phylogenetic regression[52], were conducted using the package 'geomorph'[73] in the R statistical environment[74].

**Phylogenetic tree building**. Our morphometric and evolutionary analyses require an estimate of relationships among sampled species in the form of a time-scaled phylogeny, but there is currently no single phylogenetic analysis of morphological data that has sampled all of the living and extinct anthropoid species for which we have morphometric data. As a comprehensive phylogenetic analysis of Anthropoidea is far beyond the scope of the current study, our solution to this problem

was to use the matrix representation with parsimony (MRP) approach—a method that allows multiple phylogenies with some amount of taxon overlap to be combined into a single parsimony-derived consensus tree (see ref. [75]).

For this analysis, we used MRP to combine Springer et al.'s[76] molecular phylogeny of extant primates (specifically their time-scaled tree derived from analysis using autocorrelated rates and hard bounds) with Stevens et al.'s[1] parsimony analysis of living and extinct catarrhines (their Supplementary Fig. 11A), Kay's[77] parsimony analysis of living and extinct platyrrhines, and Strait and Grine's[78] parsimony analysis of hominins. None of these studies sampled the middle Miocene hominoids *Dryopithecus* and *Hispanopithecus*, whose relationships are contentious. One or both of these closely related taxa have been interpreted at various times as stem hominids, e.g. ref. [67], stem pongines, e.g. ref. [79], stem hominines, e.g. ref. [80], or as basal crown hominoids of unresolved position relative to pongines and hominines[66]. The time-scaled tree shown in the main text (Fig. 3a) places *Dryopithecus* and *Hispanopithecus* as advanced stem hominids, more closely related to crown Hominidae than any other sampled stem hominids, but we also present 'surface' analyses in which *Dryopithecus* and *Hispanopithecus* are placed as stem pongines and as stem hominines (Supplementary Fig. 5). Importantly, these alternative placements do not affect the major results of the 'surface' analyses. Nexus trees are available in Supplementary Materials of this article.

We used the following point estimates for the extinct taxa included in the tree: *A. zeuxis*, 29.85 Ma (mean of 30.2–29.5 Ma range required by the preferred magnetostratigraphic correlation of Seiffert[28]; *Australopithecus afarensis*, 3.3 Ma (mid-way between the ~3.4 Ma age of the A.L. 129-1a and b distal femur and proximal tibia and the ~3.2 age of A.L. 288-1)[81]; *Dryopithecus fontani*, 11.9 Ma[82]; *Ekembo nyanzae*, 17.8 Ma[83]; *Equatorius*, 15.47 Ma (mean of the 15.58–15.36 Ma range provided by Behrensmeyer et al.[84]; *Hispanopithecus laietanus*, 9.6 Ma[82]; *Homo erectus*, 2 Ma (based on the maximum 1.98 Ma age for the KNM-ER 1481 specimen sampled here)[85]; *Homunculus patagonicus*, 17.2 Ma (difference between minimum and maximum ages provided by Kay[77]); *Morotopithecus bishopi*, 20.6 Ma[86]; *Paranthropus robustus*, 2 Ma (specimens sampled here, SK 87 and SK 92, are from the 'Hanging Remnant' of Swartkrans Member 1[87], which has recently been bracketed to be between ~1.8 Ma and ~2.24 in age[88]; we take the approximate midpoint of these dates); *Victoriapithecus macinnesi*, 15 Ma[89]. Divergence times among extant taxa are those in the time-scaled tree provided by Springer et al.[76], while nodes connecting extinct taxa to that tree were placed either 1 Ma older than adjacent crown nodes (if the extinct taxon is younger than that crown node; as for *Australopithecus*, *Dryopithecus*, *Equatorius*, *Hispanopithecus*, *Homunculus*, *Paranthropus*, and *Victoriapithecus*) or 1 Ma older than the taxon's geological age (if the extinct taxon is older than that crown node; as for *Aegyptopithecus*, *Homo erectus*, and *Morotopithecus*).

**Evolutionary modelling**. The most frequently used statistical model of evolution, based on its simplicity, is Brownian motion, which assumes that traits change at each unit of time with a mean change of zero and unknown and constant variance[90–92]. Within Brownian motion, the evolution of a continuous trait '*X*' along a branch over time increment '*t*' is quantified as

$$dX(t) = \sigma dB(t) \tag{1}$$

where '*σ*' constitutes the magnitude of undirected, stochastic evolution ('*σ2*' is generally presented as the Brownian rate parameter) and 'd*B*(*t*)' is Gaussian white noise. Although novel phylogenetic comparative methods continue using Brownian evolution as a baseline model, they incorporate additional parameters to model possible deviations from the pure gradual mode of evolution assumed by Brownian motion. OU models incorporate stabilizing selection as a constraint and hereby quantify the evolution of a continuous trait '*X*' as

$$dX(t) = \alpha[\theta - X(t)]dt + \sigma dB(t) \tag{2}$$

where '*σ*' captures the stochastic evolution of Brownian motion, '*α*' determines the rate of adaptive evolution towards an optimum trait value '*θ*'. When '*α*' equals zero, the deterministic part of the equation disappears and the model becomes identical to Brownian motion[43]. This standard OU model has been modified into multiple-optima OU models allowing optima to vary across the phylogeny[93]. In these implementations, the parameters are defined a priori, allowing testing of the relative likelihood of alternative parameterizations (each one characterizing a different evolutionary scenario that explains the evolution of a trait; i.e. the 'painted trees'). However, this approach leaves open the possibility that the 'best-fit' evolutionary scenario is not included in the research design. To solve this problem, the OU model fitting approach used in this study represents an extension to estimate the number of shifts and their locations on the phylogeny, rather than assuming them a priori[44]. This 'surface' method—'SURFACE Uses Regime Fitting with Akaike Information Criterion (AIC) to model Convergent Evolution'—was developed specifically to identify instances of convergent evolution and can be used to extract the evolutionary scenario with the best statistical fit (i.e. showing the lowest AICc)[45,46] between the phylogeny and the observed measurements. To avoid overfitting and to focus only in mean multivariate phenotype differences (i.e. '*θ*' the different inferred adaptive optima), 'surface' constrains both '*α*' and '*σ*'.

Hansen's 'half time' ('*t₁/₂*'; or 'phylogenetic half-life' in 'surface') was used as a proxy of rate of adaptation. It represents the time that it takes for the expected phenotype to have moved half-way to the new optimum starting in

the ancestral state[43]:

$$t_{1/2} = \ln(2)\alpha^{\wedge} - 1 \qquad (3)$$

**Reporting summary**. Further information on research design is available in the Nature Research Reporting Summary linked to this article.

## Data availability

Three-dimensional digital data from the new *Aegyptopithecus* femur described in this study (DPC 24466) are available in MorphoSource's project 'Duke Lemur Center Division of Fossil Primates' (P114), under the media number M47511. Raw data used in all geometric morphometric and evolutionary analyses are available through the 'figshare' repository at https://doi.org/10.6084/m9.figshare.9461459 [https://figshare.com/s/ebbc90c136319f9ff2cf]. Nexus trees used in the evolutionary modelling are also available as Supplementary Data 1–3. The source data underlying Figs. 2 and 3b, c, Supplementary Figs. 2 and 5, and Supplementary Table 4 are provided as a Source Data file.

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

## Acknowledgements

This study has been facilitated by the technical advice of Dean Adams and Mike Collyer ('geomorph' package) and Jeroen Smaers (valuable general advice on the phylogenetic comparative methods used in this study). We are also indebted to the following colleagues for allowing us access to high-quality casts of some of the fossil femora included in this study: Brenda Benefit and Monte McCrossin (*Victoriapithecus*; KNM-MB 35518, first appearing in ref. [10]), Laura MacLatchy (*Morotopithecus*), Salvador Moyà-Solà and David Alba (cf. *Dryopithecus* and *Hispanopithecus*). We thank Prithijit Chatrath for managing the 2009 Fayum field season, when the new *Aegyptopithecus* femur was found, and the 2009 field crew for their collecting efforts. Fieldwork in the Fayum area was facilitated by scientists from the Egyptian Geological Museum, the Egyptian Mineral Resources Authority, and the Egyptian Environmental Affairs Agency. We are also grateful to these funding sources: the Spanish Agencia Estatal de Investigación European Regional Development Fund of the European Union (CGL2017-82654-P, AEI/FEDER EU), the CERCA Programme (Generalitat de Catalunya); National Science Foundation grants BCS-0416164 and BCS-0819186 to Elwyn L. Simons and E.R.S.; and BCS-1231288 to E.R.S., Doug M. Boyer, Gregg F. Gunnell and J.G.F. This is Duke Lemur Center publication 1440 and NYCEP morphometrics contribution 111.

## Author contributions

S.A. and E.R.S. designed the study. S.A., M.T., A.S.H. and E.R.S. collected and analysed the data. All the authors discussed the results and wrote the paper.

## Competing interests

The authors declare no competing interests.
