## [Peer Review File · Nature Communications]

Reviewers' Comments:

Reviewer #1:

Remarks to the Author:

What are the major claims of the paper?

The authors apply 3D morphometrics and phylogenetic tools to a previously undescribed femur of the stem catarrhine *Aegyptopithecus* in order to elucidate patterns of similarity among anthropoid femora, and ultimately, reconstruct evolutionary trajectories of femoral anatomy through time. They find that the femur of *Aegyptopithecus* most closely resembles those of the stem platyrrhine *Homunculus* and the stem cercopithecoïd *Victoriapithecus*. The authors claim that this highlights the divergent evolutionary trajectories that cercopithecoïds and hominoids underwent in terms of their femoral anatomy.

Are the major claims of the paper novel and will they be of interest to others in the community and the wider field?

This study is very similar in methods and findings to a previous work by the same author (Almecija et al 2013). Although the focus of the 2013 paper was on the fossil hominin *Orrorin*, the shape space generated in the 2013 study is very similar to this study (see Figure 2 in Almecija et al, 2013). Although the authors in the current paper train their sights on a different time period in anthropoid femoral evolution, namely, to early catarrhine femoral evolution instead of early hominins, their findings, that *Aegyptopithecus* resembles primitive platyrrhines, and was an arboreal quadruped, are not novel. The findings corroborate previous descriptions/analyses of three *Aegyptopithecus* femora by Ankel-Simons and colleagues (1998). Thus, while suitable for a specialty journal, the paper is not sufficiently original or of sufficiently broad interest for this publication.

Is the work convincing, and if not, what further evidence would be required to strengthen the conclusions?

Ciochon and Corruccini, 1975 conducted a morphometric analysis of platyrrhine femora and found that the fossil taxa *Homunculus* and *Cebupithecia* were both most similar in their femoral anatomy to callitrichid primates. Given this, the authors should include callitrichids in future analyses.

Reviewer #2:

Remarks to the Author:

Almecija et al., NC Submission

Summary

This ms explores proximal femoral evolution in catarrhines using a combined fossil sample including primate femora from the Oligocene. They use an innovative approach - combining a new fossil description of one of the fossil linchpins of their analysis (*Aegyptopithecus*), a exploration of shape using GM and PCA, and finally a phylogenetic analysis testing hypotheses of catarrhine femoral evolution starting with early catarrhine fossil representatives. Results suggest that *Aegyptopithecus* and other early femora possessed a morphology that differed from any living representative appears to be a reasonable ancestral state from which later catarrhines and platyrrhines diverged.

I enjoyed reading this ms very much and feels it has the potential to make an exceedingly valuable contribution to early catarrhine and early hominoid evolution. As a member of a team that attempted to combine a new fossil description with a phylogenetic analysis of that fossil's implications, I know that reviewers can be quite harsh in moving the human evolution conversation towards a new approach. For that reason alone I strongly commend the authors on their innovative approach, and hope my ringing endorsement might cancel a potential negative comment from other sources. But

separately, I think the authors make a good case for their overall interpretation of early proximal femur evolution.

I also found the ms well written and quite clear, and found very few grammatical issues overall.

Main Issues

1. Phylogenetic analyses - My main concern with the ms is with the phylogenetic analysis - not the overall idea behind it, which is a fine one, but the actual realized approach. My concern comes from the use of PCs of GM shape variables in evolutionary analyses - requiring multivariate phylogenetic comparative methods. There has been some background grumbling going on for the last few years about the best way to analyze multivariate data.

The approach used in this ms is to use standard PCA to reduce shape variation into its major axis, and then use the first 3 PCs and centroid size in the main analyses. These PCs and size are run together in the R package SURFACE, which fits the most likely evolutionary model to the data using the lowest AICc scores. SURFACE assumes that each trait dimension is independent, and the final AICc scores SURFACE uses to select the best model are summary metrics (logL and AIC) from each dimension.

This approach used here is quite similar to one used recently in another primate/hominin fossil paper by Prang (2019), and numerous other articles, though the author in that case used principal components of geometric mean standardized linear variables. The issues below have, as far as I have read, not been discussed in those publications, but they should be part of the conversation as lack of previous acknowledgement is not a good excuse for this to continue.

There are two main issues with the approach the authors use in the ms, which I briefly summarize below. Each of the referenced works is more recent, but goes through the background of these issues in a more lengthy way than I have, and the authors should review them when revising their ms.

a. Multivariate issues: Adams et al. 2018 showed that using multidimensional data generated under a Brownian-Motion (BM) model in simulated SURFACE analyses led to a shifts being found in over 95% of the simulated data sets (Fig. 2c) - in other words, using multidimensional data, such as the 3 PCs and size as done in this ms, could be inappropriately finding shifts where there are none - the appropriate model is actually BM. This occurs because PC axes are correlated evolutionarily and summing the likelihood of the models across the individual dimensions leads to incorrect results (Adams et al. 2018).

b. PCA issues: Bastide et al. 2018 showed that in the presence of simulated shifts, performing PCA (and PCA corrected for phylogeny, phylogenetic PCA) on correlated traits mapped on a phylogeny led to changes in the relationships between traits - the first eigenvector was no longer in the direction of greatest variance (Fig. 1 in Bastide et al. 2018), meaning that the results of a phylogenetic analysis using PCs may be misleading. Bastide et al. (2018) presented their own approach to dealing with this problem, which allows for correlations among traits. The issue with Bastide et al. 2018 is that this approach does not work for non-ultrametric trees (i.e. those that include fossils), as used in this ms.

So where does this leave us? First, I think that issues with the approach used in the ms should be discussed in the ms. Second, the authors should run each PC and lnCS in SURFACE independently, and include the results and an interpretation of their results in the ms. If one of the main issues with using multivariate approaches is the "multivariate" part of the sentence, then run univariate analyses. I expect that the results will complement those of the multivariate approach in some interesting ways - the first PC will likely lead to regime shifts between the large clades, the second smaller, third nearer to the tips. This has been mentioned previously - Polly, 2013, but it will be interesting to view the

results and what they mean.

2. No Parameter Estimates in ms - SURFACE, like all phylogenetic comparative programs, is meant to estimate evolutionary parameters - in SURFACE, these include the phylogenetic half-life ($\ln(2)/\alpha$ - Hansen (1997)), which gives you an idea of the rate of adaptation - the average amount of time it takes to evolve half way to the new optimum given a regime shift. These should be included and discussed in the ms - are the half lives much longer than the length of the tree? Shorter? This gives an idea of how quickly femoral morphology adapts to the new optima (see Hansen, 1997 for more). These should be included and discussed biologically.

3. SI Fig 5: What is this showing us? In the main text legend of Fig. 3 it is described as "Figure S5 depicts the evolutionary history of catarrhine femoral differentiation along the first three PC axis". But the legend for SI Fig. 5 is titled "Phylogenetic sensitivity analysis for evolutionary modeling". Please be clear in the figures and the ms text.

4. I ask because the first title suggests something that should be done and interpreted for the ms - running separate analyses on each PC in SURFACE and discussing how the results compare to the main multidimensional analysis - see point 1 above.

5. Further discussion as to the meaning of the phylogenetic signal in the proximal femur, currently restricted to a few lines in one paragraph, is warranted. a) What is the biological meaning of the varying level of phylogenetic signal in the broad sense? b) What happens when you run the first 3 PCs and \ln centroid size, as was done in the SURFACE analyses in the main text? Would a finding of BM mean that the shifts found by SURFACE are less well supported? c) Finally, when you looked at the first 3 PCs vs all Procrustes coordinates (Line 314), discuss what the difference between those results (close to BM) and the overall shape via Procrustes coordinates (less phylogenetic signal than expected) actually means, and interpret this result biologically.

Minor Issues

1. Page 5, line 125: "... such as those [THAT] are used during ..."

2. Page 6, line 156: "Given the evidence presented above, plus the fact that the shaft is build more like those of cercopithecoids than hominoids (based on the interquartile range overlap in Supplementary Figure 2) ... This line refers to the shape of the DCP 24466 femur, but also Aegyptopithecus in general. But when looking at SI Fig. 2, the proportions for Aegyptopithecus look pretty similar to the range for Pan, a hominoid, and also various cercopithecoids and platyrrhines. Please rectify this conflicting statement.

3. Page 11, - heading for Supplementary Fig. S7 - I think this should be Supplementary Figure 7, or however NC has their SI format.

References

- Polly, P. D., Lawing, A. M., Fabre, A.-C. & Goswami, A. Phylogenetic Principal Components Analysis and Geometric Morphometrics. *Hystrix, the Italian Journal of Mammalogy* 24, 33–41 (2013).
- Bastide, P., Ané, C., Robin, S. & Mariadassou, M. Inference of Adaptive Shifts for Multivariate Correlated Traits. *Systematic Biology* 113, 2158–19 (2018).
- Adams, D. C. & Collyer, M. L. Multivariate Phylogenetic Comparative Methods: Evaluations, Comparisons, and Recommendations. *Syst Biol* 67, 14–31 (2018).
- Prang, T. C. The African ape-like foot of *Ardipithecus ramidus* and its implications for the origin of bipedalism. [cdn.elifesciences.org](https://doi.org/10.7554/eLife.44433) (2019). doi:10.7554/eLife.44433.001

Hansen, T. F. Stabilizing Selection and the Comparative Analysis of Adaptation. *Evolution* 51, 1341–1351 (1997).

Reviewer #3:

Remarks to the Author:

This manuscript provides important new evidence about the morphology of the proximal femur of fossil primates, especially of specimens representing stem members of the catarrhine-hominoid and platyrrhine evolutionary divergence. The authors argue convincingly that the locomotor repertoire of the early forms was wider than in extant catarrhines. Another relevant insight from this research is that ancestral forms cannot be inferred from extant morphologies but must be dug up in the field. The novelty of this information is of interest to a wide scientific audience.

line 85 ("starting point"). It might be important to state this as a hypothesis (e.g. Ae. as a model for the LCA), as we do not really know whether Ae. might have had some locomotor specializations (lines 339 and following provide a quite detailed description of its inferred locomotor habits).

line 139 ("relatively straight in all views"). What does that mean? No ap bending?

line 168 etc. (between-group PCA, Fig. 2). Between-group PCA and post-hoc projection might be problematic, as this procedure constrains perceived morphological affinities of the fossil forms according to between-group polarities that actually evolved "post-hoc". My suggestion: given the quite large fossil sample, it could be interesting to project the recent taxa into the fossil morphospace, or just perform a classical PCA of shape on all specimens (which might turn out to be very similar to Fig. 3A).

lines 216, 235, 273. (evolutionary modeling). The multi-regime OU model is fine but the authors might provide some information on potential limitations of the validity of the 10-state model result. Better statistical fit does not necessarily mean a better biological model (see overfitting problem). This might also be relevant to discuss the findings presented in the last paragraph of the results (lines 312 etc.).

I found the following paper quite helpful in providing a general perspective: Cooper, Natalie, et al. A cautionary note on the use of Ornstein Uhlenbeck models in macroevolutionary studies. *Biological Journal of the Linnean Society* 118.1 (2016): 64-77. Overall, it might be important to state that the OU model is useful to explore/characterize/categorize potential adaptive regimes/regime shifts.

line 261: "more cranially situated heads" sounds awkward, but it might be the only adequate way to characterize this morphology.

Fig. 3c: I found it hard to associate these transparent colors with the non-transparent ones in panels a,b,c.

line 316 (clear phylogenetic pattern). Neutral pattern?

line 319 (K-values). Please describe what significance means here. Significantly different from K=1?

One finding that the authors may wish to highlight/discuss at greater depth is the striking dissimilarity between Homunculus and all extant NWMs (which seems to be even bigger than the dissimilarity between Aegyptopithecus and its OW fellows). What does that mean in terms of NWM evolution?

Indeed, the similarity between Aegyptopithecus and Homunculus is important, and the significance of this finding is somewhat understated.

Responses to reviewers' comments:

In general, we are pleased with the reviewers' thoughtful and constructive comments for our study. Based on their feedback, the revised version of our work presents:

- New Figure 2 with morphospace built using both extant and fossil species (so they both contribute equally to the eigenanalysis)
- Extra Supplementary Figure (SFig5) [our previous main Figure 2 (which we believe is important)]
- Extra Supplementary Figure (SFig8) with a set of 4 different univariate 'surface' analyses (as per Reviewer 2)
- Revised methods, providing more details (especially the potential pitfalls) of OU models
- Revised results section: including a new morphospace (including fossils in the eigenanalysis), extra OU (univariate) modelling, phylogenetic half-lives of each variable, and extended phylogenetic signal analyses
- Revised discussion of the new results
- All minor suggested edits were implemented

Below we detail how we answered each reviewer comment (in bold):

Reviewer #1 (Remarks to the Author):

What are the major claims of the paper?

The authors apply 3D morphometrics and phylogenetic tools to a previously undescribed femur of the stem catarrhine *Aegyptopithecus* in order to elucidate patterns of similarity among anthropoid femora, and ultimately, reconstruct evolutionary trajectories of femoral anatomy through time. They find that the femur of *Aegyptopithecus* most closely resembles those of the stem platyrrhine *Homunculus* and the stem cercopithecoid *Victoriapithecus*. The authors claim that this highlights the divergent evolutionary trajectories that cercopithecoids and hominoids underwent in terms of their femoral anatomy.

Are the major claims of the paper novel and will they be of interest to others in the community and the wider field?

This study is very similar in methods and findings to a previous work by the same author (Almecija et al 2013). Although the focus of the 2013 paper was on the fossil hominin *Orrorin*, the shape space generated in the 2013 study is very similar to this study (see Figure 2 in Almecija et al, 2013). Although the authors in the current paper train their sights on a different time period in anthropoid femoral evolution, namely, to early catarrhine femoral evolution instead of early hominins, their findings, that *Aegyptopithecus* resembles primitive platyrrhines, and was an arboreal quadruped, are not novel. The findings corroborate previous descriptions/analyses of three *Aegyptopithecus* femora by Ankel-Simons and colleagues (1998). Thus, while suitable for a specialty journal, the paper is not sufficiently original or of sufficiently broad interest for this publication.

Reviewer 1 states that “their findings, that *Aegyptopithecus* resembles primitive platyrrhines, and was an arboreal quadruped, are not novel”. We consider this summary of our study’s findings to be overly simplistic and dismissive. The analytical approach that we have taken in the current study derives from our observations on the new *Aegyptopithecus* femoral specimen (DPC 24466), which

shows for the first time that the proximal femoral anatomy of this important stem catarrhine does not resemble that of either extant hominoids or extant cercopithecoids; this, in itself, is an entirely novel finding. We must emphasize that paleontologists have been working in the early Oligocene beds of the Fayum area for well over a century, and despite the recovery of hundreds of anthropoid fossils, details of the proximal femoral morphology of this most famous Fayum anthropoid – *Aegyptopithecus* -- have not been known until now. Given that the proximal femur plays a key role in interpretations of locomotor behavior in Miocene catarrhines, DPC 24466 is a clearly a critically important addition to our understanding of the body plan of advanced stem catarrhines. Adding *Aegyptopithecus* to the morphometric data set in turn reveals that this advanced stem catarrhine is intermediate in morphology between cercopithecoids and hominoids; this, too, is an entirely novel finding. We do find that *Aegyptopithecus* is similar in morphology to an early platyrrhine (*Homunculus*) – another novel finding – but contrary to expectation, *Homunculus* is found to be more similar to stem catarrhines than to crown platyrrhines in proximal femoral morphology (yet another novel finding). Our analyses further provide strong evidence that the newly observable proximal femoral morphology of *Aegyptopithecus*, and the previously observable but difficult to interpret proximal femoral morphology of *Homunculus*, likely represent the ancestral condition for crown anthropoids, and that crown platyrrhines have diverged significantly from this ancestral morphology (another, very important, novel finding). It is not surprising that the proximal femoral anatomy of *Aegyptopithecus* is consistent with it having been an arboreal quadruped; this inference has not changed since the 1970s as evidence has accumulated from other parts of *Aegyptopithecus*' skeletal anatomy. But confirming this locomotor inference is only a minor aspect of the current study – much more important are the novel observations listed above, showing that *Aegyptopithecus*' proximal femoral morphology is unlike that of any living catarrhine or platyrrhine, and yet likely represents the ancestral condition for all living anthropoids. Finally, another novel aspect of the current study relative to that of Almécija et al. 2013 is that this study incorporates a much more complex and thorough evolutionary modeling component (i.e., multivariate multi-optima Ornstein-Uhlenbeck as compared to more traditional maximum likelihood methods assuming Brownian motion).

Is the work convincing, and if not, what further evidence would be required to strengthen the conclusions?

Ciochon and Corruccini, 1975 conducted a morphometric analysis of platyrrhine femora and found that the fossil taxa *Homunculus* and *Cebupithecia* were both most similar in their femoral anatomy to callitrichid primates. Given this, the authors should include callitrichids in future analyses.

There are clear differences between the Ciochon and Corruccini study and the present one. For example, the former relied on linear measurements from all over the femur (including also the distal portion), and *only* sampled platyrrhines. There is no reason to assume that the morphospaces of both studies (and conclusions driven from them) should be comparable. While it could be interesting to sample callitrichids to test whether they retained the primitive platyrrhine and anthropoid morphology, the fact that crown Callitrichidae is a highly derived and deeply nested clade that arose relatively recently (middle or late Miocene) when compared to the very ancient *Aegyptopithecus*, their inclusion is highly unlikely to have any impact on the primary results and

conclusions of our study. In light of this, and given that adding callitrichids to our already huge data set (N = 503 extant, 14 fossils) would significantly delay our resubmission, we have decided that this request is beyond the scope of the current study.

Reviewer #2 (Remarks to the Author):

Almécija et al., NC Submission

Summary

This ms explores proximal femoral evolution in catarrhines using a combined fossil sample including primate femora from the Oligocene. They use an innovative approach - combining a new fossil description of one of the fossil linchpins of their analysis (*Aegyptopithecus*), a exploration of shape using GM and PCA, and finally a phylogenetic analysis testing hypotheses of catarrhine femoral evolution starting with early catarrhine fossil representatives. Results suggest that *Aegyptopithecus* and other early femora possessed a morphology that differed from any living representative appears to be a reasonable ancestral state from which later catarrhines and platyrrhines diverged.

I enjoyed reading this ms very much and feels it has the potential to make an exceedingly valuable contribution to early catarrhine and early hominoid evolution. As a member of a team that attempted to combine a new fossil description with a phylogenetic analysis of that fossil's implications, I know that reviewers can be quite harsh in moving the human evolution conversation towards a new approach. For that reason alone I strongly commend the authors on their innovative approach, and hope my ringing endorsement might cancel a potential negative comment from other sources. But separately, I think the authors make a good case for their overall interpretation of early proximal femur evolution.

I also found the ms well written and quite clear, and found very few grammatical issues overall.

Main Issues

1. Phylogenetic analyses - My main concern with the ms is with the phylogenetic analysis - not the overall idea behind it, which is a fine one, but the actual realized approach. My concern comes from the use of PCs of GM shape variables in evolutionary analyses - requiring multivariate phylogenetic comparative methods. There has been some background grumbling going on for the last few years about the best way to analyze multivariate data.

The approach used in this ms is to use standard PCA to reduce shape variation into its major axis, and then use the first 3 PCs and centroid size in the main analyses. These PCs and size are run together in the R package SURFACE, which fits the most likely evolutionary model to the data using the lowest AICc scores. SURFACE assumes that each trait dimension is independent, and the final AICc scores SURFACE uses to select the best model are summary metrics (logL and AIC) from each dimension.

This approach used here is quite similar to one used recently in another primate/hominin fossil paper by Prang (2019), and numerous other articles, though the author in that case used principal components of geometric mean standardized linear variables. The issues below have, as far as I have read, not been discussed in those publications, but they should be part of the conversation as lack of previous acknowledgement is not a good excuse for this to continue.

There are two main issues with the approach the authors use in the ms, which I briefly summarize below. Each of the referenced works is more recent, but goes through the background of these issues in a more lengthy way than I have, and the authors should review them when revising their ms.

a. Multivariate issues: Adams et al. 2018 showed that using multidimensional data generated under a Brownian-Motion (BM) model in simulated SURFACE analyses led to a shifts being found in over 95% of the simulated data sets (Fig. 2c) - in other words, using multidimensional data, such as the 3 PCs and size as done in this ms, could be inappropriately finding shifts where there are none - the appropriate model is actually BM. This occurs because PC axes are correlated evolutionarily and summing the likelihood of the models across the individual dimensions leads to incorrect results (Adams et al. 2018).

b. PCA issues: Bastide et al. 2018 showed that in the presence of simulated shifts, performing PCA (and PCA corrected for phylogeny, phylogenetic PCA) on correlated traits mapped on a phylogeny led to changes in the relationships between traits - the first eigenvector was no longer in the direction of greatest variance (Fig. 1 in Bastide et al. 2018), meaning that the results of a phylogenetic analysis using PCs may be misleading. Bastide et al. (2018) presented their own approach to dealing with this problem, which allows for correlations among traits. The issue with Bastide et al. 2018 is that this approach does not work for non-ultrametric trees (i.e. those that include fossils), as used in this ms.

So where does this leave us? First, I think that issues with the approach used in the ms should be discussed in the ms. Second, the authors should run each PC and lnCS in SURFACE independently, and include the results and an interpretation of their results in the ms. If one of the main issues with using multivariate approaches is the “multivariate” part of the sentence, then run univariate analyses. I expect that the results will complement those of the multivariate approach in some interesting ways - the first PC will likely lead to regime shifts between the large clades, the second smaller, third nearer to the tips. This has been mentioned previously - Polly, 2013, but it will be interesting to view the results and what they mean.

We appreciate the positive feedback and thoughtful constructive discussion that Reviewer 2 has provided. The main concern of Reviewer 2 relates to the findings of Adams and Collyer (2018, in *Systematic Biology*) who claimed that “methods that summarize patters across trait dimensions treated separately (e.g., *SURFACE*) incorrectly assume independence among trait dimensions,” which can result in model misspecification. Bastide et al.’s (2018, also in *Systematic Biology*) concerns also relate to correlated traits. Therefore, to address this key potential issue, we tested if the different principal components included in the evolutionary analyses (PCs 1-3) were correlated with each other using pGLS. The results indicate that although PC1 is correlated with PC2 and PC3 (but not PC2-PC3), the relationship explains only a very small portion of the predictable covariation (R^2 PC1-2 = 0.009801, R^2 PC1-3 = 0.002116). In light of this, we are confident that our multivariate multi-OU modeling is robust. However, following the advice of reviewer2, we complement our results and discussion with the univariate fits of each variable (Fig. S8). However, we also note that SURFACE performs much better using 2 to 4 variables at least (Ingram and Mahler 2013 *MethEcolEvol*). Furthermore, following also the advice of Reviewer 2, we have added a discussion of the pitfalls associated with multivariate OU modeling, and appropriate references.

2. No Parameter Estimates in ms - SURFACE, like all phylogenetic comparative programs, is meant to estimate evolutionary parameters - in SURFACE, these include the phylogenetic half-life ($\ln(2)/\alpha$ - Hansen (1997)), which gives you an idea of the rate of adaptation - the average amount of time it takes to evolve half way to the new optimum given a regime shift. These should be included and discussed in the ms - are the half lives much longer than the length of the tree? Shorter? This gives an idea of how quickly femoral morphology adapts to the new optima (see Hansen, 1997 for more). These should be included and discussed biologically.

This is very interesting information that we did not include in the original version for the sake of space and simplicity. However, we agree that these are important, and $t_{1/2}$ estimates of PC1, PC2, PC3 and lnCS are now provided and discussed in the main text. Interestingly, these results indicate that, amongst others, the changes related to the divergent pattern between hominoids and cercopithecoids (as captured in PC1) occurred twice as fast as that between platyrrhines and catarrhines (as captured by PC2).

3. SI Fig 5: What is this showing us? In the main text legend of Fig. 3 it is described as “Figure S5 depicts the evolutionary history of catarrhine femoral differentiation along the first three PC axis”. But the legend for SI Fig. 5 is titled “Phylogenetic sensitivity analysis for evolutionary modeling”. Please be clear in the figures and the ms text.

We meant Supplementary Figure 4. It has been emended in the caption of Figure 3.

4. I ask because the first title suggests something that should be done and interpreted for the ms - running separate analyses on each PC in SURFACE and discussing how the results compare to the main multidimensional analysis - see point 1 above.

Change applied, see above.

5. Further discussion as to the meaning of the phylogenetic signal in the proximal femur, currently restricted to a few lines in one paragraph, is warranted. a) What is the biological meaning of the varying level of phylogenetic signal in the broad sense? b) What happens when you run the first 3 PCs and ln centroid size, as was done in the SURFACE analyses in the main text? Would a finding of BM mean that the shifts found by SURFACE are less well supported? c) Finally, when you looked at the first 3 PCs vs all Procrustes coordinates (Line 314), discuss what the difference between those results (close to BM) and the overall shape via Procrustes coordinates (less phylogenetic signal than expected) actually means, and interpret this result biologically.

Following the advice of Reviewer 2, the phylogenetic signal for PC1-PC3 plus lnCS was also added ($K = 2.245$), and a paragraph in the discussion was added, as an attempt to provide a biological interpretation of these results. Specifically, we hypothesize that only a few principal components (PC1-3, see our Supplementary Fig. 3) capture meaningful shape information and that lnCS (in this case a proxy femoral size) is, not surprisingly, tracking changes in body size.

Minor Issues

1. Page 5, line 125: "... such as those [THAT] are used during ..."
2. Page 6, line 156: "Given the evidence presented above, plus the fact that the shaft is build more like those of cercopithecoids than hominoids (based on the interquartile range overlap in Supplementary Figure 2) ... This line refers to the shape of the DCP 24466 femur, but also Aegyptopithecus in general. But when looking at SI Fig. 2, the proportions for Aegyptopithecus look pretty similar to the range for Pan, a hominoid, and also various cercopithecoids and platyrrhines. Please rectify this conflicting statement.
3. Page 11, - heading for Supplementary Fig. S7 - I think this should be Supplementary Figure 7, or however NC has their SI format.

All minor changes were applied (sentence in point 2 was misplaced and now deleted from this context).

References

- Polly, P. D., Lawing, A. M., Fabre, A.-C. & Goswami, A. Phylogenetic Principal Components Analysis and Geometric Morphometrics. *Hystrix, the Italian Journal of Mammalogy* 24, 33–41 (2013).
- Bastide, P., Ané, C., Robin, S. & Mariadassou, M. Inference of Adaptive Shifts for Multivariate Correlated Traits. *Systematic Biology* 113, 2158–19 (2018).
- Adams, D. C. & Collyer, M. L. Multivariate Phylogenetic Comparative Methods: Evaluations, Comparisons, and Recommendations. *Syst Biol* 67, 14–31 (2018).
- Prang, T. C. The African ape-like foot of *Ardipithecus ramidus* and its implications for the origin of bipedalism. cdn.elifesciences.org (2019). doi:10.7554/eLife.44433.001
- Hansen, T. F. Stabilizing Selection and the Comparative Analysis of Adaptation. *Evolution* 51, 1341–1351 (1997).

Reviewer #3 (Remarks to the Author):

This manuscript provides important new evidence about the morphology of the proximal femur of fossil primates, especially of specimens representing stem members of the catarrhine-hominoid and platyrrhine evolutionary divergence. The authors argue convincingly that the locomotor repertoire of the early forms was wider than in extant catarrhines. Another relevant insight from this research is that ancestral forms cannot be inferred from extant morphologies but must be dug up in the field. The novelty of this information is of interest to a wide scientific audience.

We really appreciate the vision of Reviewer 3, we could not have expressed it better.

line 85 ("starting point"). It might be important to state this as a hypothesis (e.g. Ae. as a model for the LCA), as we do not really know whether Ae. might have had some locomotor specializations (lines 339 and following provide a quite detailed description of its inferred locomotor habits).

Revised text has been rephrased to clarify our assumptions: “Given that *Aegyptopithecus* is close in age to the predicted divergence of cercopithecoids and hominoids, and is widely accepted as an advanced stem catarrhine, we proceed with the assumption that this taxon is more likely than not to closely approximate the morphology of the last common ancestor of cercopithecoids and hominoids, and is not already highly autapomorphic.”

line 139 (“relatively straight in all views”). What does that mean? No ap bending?

Exactly. The lateral and medial views of the specimen in Fig. 1b shows that there is no evident ap bending.

line 168 etc. (between-group PCA, Fig. 2). Between-group PCA and post-hoc projection might be problematic, as this procedure constrains perceived morphological affinities of the fossil forms according to between-group polarities that actually evolved “post-hoc”. My suggestion: given the quite large fossil sample, it could be interesting to project the recent taxa into the fossil morphospace, or just perform a classical PCA of shape on all specimens (which might turn out to be very similar to Fig. 3A).

We truly appreciate the insight from Reviewer 3 regarding the “fossil-only bgPCA”. As an experiment, we performed this analysis and found that by maximizing the variation among fossils (that really are very similar to each other), when extant individuals are potted *post hoc* it results in a artefactual overlap of the major clades (see below). A better approach, also based on Reviewer 3’s comment, is to use the morphospace used for the evolutionary modeling (Fig. 3A), in which both living and fossil species contributed equally to the eigenanalysis, and then plot all individuals *post hoc*. This way fossils are not constrained to an extant-only morphospace. This analysis is the basis of the new Figure 2, which is similar to the former but with broader spreads of the major living clades and the fossils.

lines 216, 235, 273. (evolutionary modeling). The multi-regime OU model is fine but the authors might provide some information on potential limitations of the validity of the 10-state model result. Better statistical fit does not necessarily mean a better biological model (see overfitting problem). This might also be relevant to discuss the findings presented in the last paragraph of the results (lines 312 etc.).

I found the following paper quite helpful in providing a general perspective: Cooper, Natalie, et al. A cautionary note on the use of Ornstein Uhlenbeck models in macroevolutionary studies. *Biological Journal of the Linnean Society* 118.1 (2016): 64-77. Overall, it might be important to state that the OU model is useful to explore/characterize/categorize potential adaptive regimes/regime shifts.

Based on these comments and similar concerns from Reviewer 2, the revised version of the main text includes details about possible pitfalls of the method, with the relevant references.

line 261: “more cranially situated heads” sounds awkward, but it might be the only adequate way to characterize this morphology.

The revised text reads “more proximally situated heads.”

Fig. 3c: I found it hard to associate these transparent colors with the non-transparent ones in panels a,b,c. **The color transparency of the panels has been edited to facilitate comparisons.**

line 316 (clear phylogenetic pattern). Neutral pattern?

We substituted the wording to “Brownian phylogenetic signal” in the revised version of this text (now in a different paragraph in the Discussion).

line 319 (K-values). Please describe what significance means here. Significantly different from $K=1$?

Adams 2014 (*Systematic Biology*) explains that K_{mult} (Blomberg’s K version for multivariate data, the one implemented in this study) is evaluated statistically via permutation, where data at the tips of the phylogeny are randomized relative to the tree, and values of K_{rand} are obtained for each permutation of the data which are then compared with K_{mult} . This basically means the value K value obtained is not random. These details are clarified in the revised text.

One finding that the authors may wish to highlight/discuss at greater depth is the striking dissimilarity between *Homunculus* and all extant NWMs (which seems to be even bigger than the dissimilarity between *Aegyptopithecus* and its OW fellows). What does that mean in terms of NWM evolution? Indeed, the similarity between *Aegyptopithecus* and *Homunculus* is important, and the significance of this finding is somewhat understated.

As there is very little evidence available from *Homunculus*, we chose to keep our conclusions fairly general: **“As all crown platyrrhines occupy a regime that is different from that of the stem platyrrhine *Homunculus*, we infer that a locomotor shift likely occurred along the terminal part of the platyrrhine stem lineage (in the late Oligocene or perhaps early Miocene) from the ancestral anthropoid regime retained by *Homunculus* to one that is occupied today by *Aotus* and *Callicebus*.”**

Reviewers' Comments:

Reviewer #2:

Remarks to the Author:

Almécija et al. NC Review

09.02.2019

Remarks to the author

This is a revision of a ms I previously reviewed. The authors addressed all my concerns adequately and I appreciate the time they took in running the new analyses and incorporating them into their revision. I have only a few minor points which I outline below - these are important though, and should be addressed in a revision. I think it will make a valuable contribution to the field and could provide a new goalposts for fossil descriptions and analyses.

Minor Points - words in brackets are additions

1. Starting at line 265: Make sure the language for the "rate of adaptation" section given by the half-life estimate is correct. This parameter does not give you the rate of evolution, it is specifically the rate of adaptation towards optima given a regime shift. Some minor rewording is required in this section and in the discussion making sure this is always about the rate of adaptation, not the rate of evolution.

For example, on line 273, this sentence should be reworded in a way similar to this: " These results indicate, among other things, that [femoral morphological adaptations that distinguish hominoids and cercopithecoids] ..."

Another example: On line 374: ... based on the $t_{1/2}$ results, adapted at a faster rate to the new selective regime than the rate of adaptation that separated catarrhines and platyrrhines.

There are a few other lines that need to be reworded in a similar way.

2. Line 50: You have plesiomorphic on line 46 outside the parentheses, followed by the definition (i.e. primitive), but on line 60 you have "specialized" outside the parentheses with (i.e. autamorphic) inside the parentheses. Choose one format and stick to it.

3. Line 99: "towards neither [of] the distinct locomotor ..."

4. Line 184: "and hominoids [are] separated from ... "

5. Line 264: One of the evolutionary parameters estimated [in OU models and specifically] by 'surface ..."

6. Line 408: This study provides [new and novel] information about the evolution of ...

Reviewer #3:

Remarks to the Author:

The authors have addressed all points raised in my first review.

Here are some minor points:

line 124: typo ("in MorphoSource's")

lines 178/179: "...individual extant specimen scores were then plotted into the morphospace post-hoc". Technically, it might be more adequate to say "extant specimens were projected into the morphospace..." (because the scores are the outcome of the projection)

line 225: "two morphospaces": rather, these are two representations (subspaces) of the same

morphospace

lines 228/229: "Note that the difference between the morphospaces depicted in Figures 2 and 3b is only visual:". I would suggest a positive phrasing here, something in the sense of "Note that individual data (Fig. 2) and species means/adaptive optima (Fig. 3b) are represented in the same morphospace"

line 327: typo ("inform")

Christoph P. E. Zollikofer

REVIEWERS' COMMENTS:

[author's comments in bold]

We are delighted with the positive and constructive comments of these two reviewers. All their suggestions were implemented in the revised text (see below).

Reviewer #2 (Remarks to the Author):

Almécija et al. NC Review

09.02.2019

Remarks to the author

This is a revision of a ms I previously reviewed. The authors addressed all my concerns adequately and I appreciate the time they took in running the new analyses and incorporating them into their revision. I have only a few minor points which I outline below - these are important though, and should be addressed in a revision. I think it will make a valuable contribution to the field and could provide a new goalposts for fossil descriptions and analyses.

Minor Points - words in brackets are additions

1. Starting at line 265: Make sure the language for the “rate of adaptation” section given by the half-life estimate is correct. This parameter does not give you the rate of evolution, it is specifically the rate of adaptation towards optima given a regime shift. Some minor rewording is required in this section and in the discussion making sure this is always about the rate of adaptation, not the rate of evolution.

We agree with this definition, and we would like to clarify that we did not define the phylogenetic half-life as a “rate of evolution” parameter, but as a “rate of adaptation estimate,” and followed with a description that matches the one provided by the reviewer above.

For example, on line 273, this sentence should be reworded in a way similar to this: “ These results indicate, among other things, that [femoral morphological adaptations that distinguish hominoids and cercopithecoids] ...”

The text edit suggestion was applied.

Another example: On line 374: ... based on the $t_{1/2}$ results, adapted at a faster rate to the new selective regime than the rate of adaptation that separated catarrhines and platyrrhines.

The text edit suggestion was applied.

There are a few other lines that need to be reworded in a similar way.

2. Line 50: You have plesiomorphic on line 46 outside the parentheses, followed by the definition (i.e. primitive), but on line 60 you have “specialized” outside the parentheses with (i.e. autamorphic) inside the parentheses. Choose one format and stick to it.
 3. Line 99: “towards neither [of] the distinct locomotor ...”
 4. Line 184: “and hominoids [are] separated from ...”
 5. Line 264: One of the evolutionary parameters estimated [in OU models and specifically] by ‘surface ...’
 6. Line 408: This study provides [new and novel] information about the evolution of ...
- These minor edit suggestions were applied. The last suggestion was changed from “interesting” to “new and interesting” (because “new” and “novel” are synonymous).**

Reviewer #3 (Remarks to the Author):

The authors have addressed all points raised in my first review.

Here are some minor points:

line 124: typo (“in MorphoSource's”)

lines 178/179: “...individual extant specimen scores were then plotted into the morphospace post-hoc”. Technically, it might be more adequate to say “extant specimens were projected into the morphospace...” (because the scores are the outcome of the projection)

line 225: “two morphospaces”: rather, these are two representations (subspaces) of the same morphospace

lines 228/229: “Note that the difference between the morphospaces depicted in Figures 2 and 3b is only visual:”. I would suggest a positive phrasing here, something in the sense of “Note that individual data (Fig. 2) and species means/adaptive optima (Fig. 3b) are represented in the same morphospace”

line 327: typo (“inform”)

These minor text edits suggestions were all applied.

Christoph P. E. Zollikofer